# Fabrication and Characterization of Bioresorbable Drug-coated Porous Scaffolds for Vascular Tissue Engineering

**DOI:** 10.3390/ma12091438

**Published:** 2019-05-02

**Authors:** Jueun Kim, Su A. Park, Jei Kim, Jaejong Lee

**Affiliations:** 1Department of Nano Manufacturing Technology, Nano-Convergence Mechanical Systems Research Division, Korea Institute of Machinery & Materials (KIMM), 156 Gajeongbuk-Ro, Yuseong-Gu, Daejeon 34103, Korea; jekim@kimm.re.kr; 2Department of Nature-inspired Nano Convergence Systems, Nano-Convergence Mechanical Systems Research Division, Korea Institute of Machinery & Materials (KIMM), 156 Gajeongbuk-Ro, Yuseong-Gu, Daejeon 34103, Korea; psa@kimm.re.kr; 3Department of Neurology, Chungnam National University Hospital, 282 Munhwa-ro, Jung-gu, Daejeon 35015, Korea; jeikim@cnu.ac.kr

**Keywords:** bioresorbable polymers, polycaprolactone (PCL), vascular scaffolds, aspirin, atorvastatin calcium salt

## Abstract

Bioresorbable polymers have been studied for several decades as attractive candidates for promoting the advancement of medical science and bio-technology in modern society. In particular, with a well-defined architecture, bioresorbable polymers have prominent advantages over their bulk counterparts for applications in biomedical and implant devices, such as cell delivery, scaffolds for tissue engineering, and hydrogels as well as in the pharmaceutical fields. Biocompatible implant devices based on bioresorbable materials (for instance, bioresorbable polymers that combine the unique advantages of biocompability and easy handling) have emerged as a highly active field due to their promising applications in artificial implant systems and biomedical devices. In this paper, we report an approach to fabricate porous polycaprolactone (PCL) scaffolds using a 3D printing system. And its surface was treated to a hydrophilic surface using plasma treatment. Then, the aspirin and atorvastatin calcium salt mixture was dip coated onto the surface. The drug coating technology was used to deposit the drug material onto the scaffold surface. Our porous PCL scaffold was coated with aspirin and atorvastatin calcium salt to reduce the blood LDL cholesterol and restenosis. These results suggest that our approach may provide a promising scaffold for developing bioresorbable drug-delivery-biomaterials. We further demonstrate that our bioresorbable medical device can be used as vascular scaffolds to provide a wide range of applications for the design of medical devices.

## 1. Introduction

Synthetic bioresorbable polymers as design materials for medical applications have been reported by many researchers [1,2,3,4,5]. Bioresorbable polymers have been chosen as the optimal carrier for local drug delivery and are widely used in tissue engineering applications as scaffolds [6,7,8]. The interest in these applications continues to increase as more biocompatible polymers become available. Specifically, bioresorbable polymers are applicable to those medical devices in which tissue repair or remodeling is the goal, but not in applications for which long-term material stability is required. Typical products for these kinds of biomaterials are artificial skin, bone grafts, absorbable sutures, and stent [6,9,10,11].

The majority of synthetic bioresorbable polymers that are currently available are poly (α-hydroxyacid) based polymers that have repeating units of –O–R–CO–(R; aliphatic) in the main chain. Among the synthetic bioresorbable polymers, aliphatic polyesters such as poly-(glycolic acid or glycolide) (PGA), poly-lactic-L-lactide (PLLA), poly-lactic-D-lactide (PDLA), and polycaprolactone (PCL) are the most commonly adopted [3,6,12]. These polymers are obtained through a polycondensation reaction involving one or more different monomers. It takes longer than one year, and usually a few years, for synthetic polymers consisting of PLLA, PDLA and PCL to be completely bioabsorbed [13].

PCL is one of the most elastic among the synthetic bioresorbable polymers [14,15,16]. It is used in clinics as a surgical material, and its biomedical applications have been researched in controlled drug release applications and in long-term implants for vascular surgery [17,18,19]. PCL has a low glass transition temperature of about −60 °C, a low melting point of around 60 °C, and a high thermal stability. Because of its low glass transition temperature, the PCL amorphous phase has a high molecular mobility at human body temperature [20]. Furthermore, with its significant degree of crystallinity and hydrophobicity, high molecular weight PCL has a long-term in vivo implantable time. In addition, PCL is the U.S. Food and Drug Administration (FDA) approved as a biocompatible material, and in vitro and in vivo experiments are widely known [21]. As an aliphatic polyester, PCL also degrades by hydrolysis, and the hydrolyzed products are absorbed by the body with minimal tissue reactions. The complete decomposition of the polymer could occur within two years depending on the degree of crystallinity [6,22].

Synthetic bioresorbable polymers are commonly used in biomedical applications [23,24,25,26,27]. Specifically, bioabsorbable vascular scaffolds have been designed to overcome the long-term complications of metallic drug eluting stents (DES) to treat arterial restenosis [28]. Significant complications in the past have been found due to metallic stents such as vascular stimulation and delayed endothelial cell proliferation [29]. To address this challenge, researchers have focused on bioresorbable stents over time. Current investigation aims have attempted to develop fully bioabsorbable, drug-coated stents using synthetic bioresorbable polymers by solution coating methods such as layer by layer [30], spray coating [31] and dip coating [32]. In this study, we used the well-known bioresorbable polymer PCL with a 3D printing system to fabricate vascular scaffolds [33]. Then, we coated them with aspirin and atorvastatin calcium salt to reduce the blood LDL cholesterol and restenosis [34]. A drug coating technology was used to deposit the drug material onto the scaffold surface.

## 2. Materials and Methods

### 2.1. Materials

All the chemicals were purchased from Sigma-Aldrich. Polycaprolactone (PCL, St. Louis, MO, USA), and a mean molecular weight of 45 kDa was used as the base polymer biomaterial for the scaffolds. The PCL has a melting point of 60 °C and a glass transition temperature of −60 °C according to the manufacturer’s reports. Aspirin (ASP, Sigma-Aldrich, A2093, meeting the USP testing specifications, Seoul, Korea) and atorvastatin calcium salt (ACT, Sigma-Aldrich, ≥98%, HPLC, Seoul, Korea) were used as the drug materials (Figure 1). Anhydrous methanol and dimethyl sulfoxide (DMSO, Sigma-Aldrich, D814, Seoul, Korea) were purchased from Sigma-Aldrich.

### 2.2. Fabrication of the Porous PCL Scaffolds by 3D printing

PCL was dissolved in DMSO solution in a vial with physical mixing at 60 °C for 1 h. The PCL/DMSO mixture with the melted PCL in DMSO solution (PCL:DMSO = 60:40 volume ratio) was used in the 3D printing process. The mixture was put in the stein-less barrel and ejected through the nozzle under 60 °C. Then, the scaffolds were fabricated using a 3D printing system (laboratory-made system at the Korea Institute of Machinery and Materials). The 3D printing system made a mixture layer on the printing bed, and then, we removed the DMSO by rinsing with deionized water. The remaining DMSO in the scaffold was sufficiently washed out, and the scaffolds were dried for 24 h at room temperature.

### 2.3. Preparation of the Atorvastatin Calcium Salt: Aspirin Stock Solutions

An atorvastatin calcium salt: aspirin (ACT:ASP) solution was made by dissolving both the ACT and ASP powders into a mixture of anhydrous methanol and deionized water (1:1 volume ratio) at a molar ratio of 1:20 with stirring for 1 h at room temperature, as a stock solution (10,000 ppm).

### 2.4. Drug Coating on the Porous PCL Scaffolds

First, the porous PCL scaffolds were treated with oxygen plasma using LF plasma (Batch Type Plasma for R&D, Femto-Science Inc., Deajeon, Korea). The surface characteristics of the porous PCL scaffolds were also easily changed with oxygen to convert their surfaces from a hydrophobic to a hydrophilic surface [35,36] (Appendix A). To fabricate the drug-coated the porous PCL scaffolds, we used a stock solution and a dip coating method to apply coatings to the porous PCL scaffold. The porous PCL scaffold was immersed in the stock solution using an ultrasonic homogenizer (Ultrasonic Cleaner, Daeil Scientific Co., Ltd., Deajeon, Korea) for 10 min (Scheme 1). The drug-coated porous PCL scaffolds were vacuum-dried for 24 h to remove the solvent.

### 2.5. Characterization

#### 2.5.1. Scanning Electron Microscopy (SEM)

The morphology of the porous PCL scaffolds and the drug-coated porous PCL scaffolds were studied by scanning electron microscopy (SEM) in a S-4800 Hitachi microscope (Tokyo, Japan) using an accelerating voltage of 10 kV. Samples were first Pt coated before the imaging.

#### 2.5.2. Optical Microscopy (OM)

Optical Microscopy (OM) image of the porous PCL scaffold was obtained on an Olympus BX-51 (Tokyo, Japan).

#### 2.5.3. Fourier Transform Infrared Spectroscopy (FT-IR)

Structural characterization of the porous PCL scaffold, the drug-coated porous PCL scaffold, and the drug-released porous PCL scaffold were carried out by Fourier Transform Infrared Spectroscopy (FT-IR) using a Bruker (VERTEX 80 & 80v) FT-IR spectrometer equipped with a vacuum (Billerica, MA USA). The spectra were recorded at ambient temperature between 400 and 4000 cm^−1^.

#### 2.5.4. X-ray Photoelectron Spectroscopy (XPS)

The chemical bonding states and atomic concentrations in the samples before and after the drug coating on the porous PCL scaffold were examined by X-ray Photoelectron Spectroscopy (XPS) (K-Alpha^+^ XPS Spectrometer, Thermo Scientific, Tokyo, Japan) using a hemispherical electrostatic energy analyzer and an Al K_α_ X-ray source. The base pressure in the sample chamber was controlled to 10^−9^ Torr. The measured spectra were displayed as plots of the number of electrons versus the electron binding at a fixed, small energy interval. Peak area and peak height sensitivity factors were used for quantification.

#### 2.5.5. Drug Content

The drug content of the atorvastatin calcium salt:aspirin (1:20 molar ratio) coated porous PCL scaffold was estimated by High Performance Liquid Chromatograph (HPLC) (Agilent 1260 Infinity, Seoul, Korea). The HPLC system used for the drug analysis was an Agilent 1260 equipped with a UV visible detector. The column used for the analysis was a C18. The atorvastatin calcium salt:aspirin (1:20 molar ratio) coated porous PCL scaffold was analyzed using a mobile phase consisting of deionized water:MeOH (1:1 volume ratio) at a flow rate of 1.0 mL/min. The detector wavelength was set at 240 nm. To investigate the drug content, the atorvastatin calcium salt:aspirin (1:20 molar ratio) coated porous PCL scaffold was dissolved in 10 mL of the mobile phase, and 10 μL was injected into the HPLC system.

## 3. Results and Discussion

In this study, aspirin and atorvastatin calcium salt were imbedded onto the hydrophilic surface of a porous PCL scaffold by dip coating with an atorvastatin calcium salt: aspirin stock solution. After the drug coating, the solvent was removed by vacuum drying for 24 h, and the drug-coated porous PCL scaffolds were finally obtained.

An optical image of the porous PCL scaffold (drug-uncoated) is shown in Figure 2a. The porous morphology was selected for the preparation of the drug coating on the PCL scaffolds. The purpose of this study was to evaluate the affectivity of the facile drug coating on the scaffolds. For further analysis of the structure and surface topography of the porous PCL scaffold (drug-uncoated), we performed scanning electron microscopy (SEM) at different magnifications. The SEM images are shown in Figure 2b,c. A porous filament was observed on the PCL scaffold fabricated with the 3D printing process.

In Figure 3, the surface of the drug-coated porous PCL scaffold had a similar pristine surface morphology. This result shows that the strut surface is dependent on the solvent of the drug solution. When a methanol-based drug mixture is coated onto PCL, a phenomenon occurs in which the PCL decomposes in the methanol in a few minutes. To solve this problem, water was added to inhibit the decomposition of the PCL. However, when the ratio of water in the mixed solvent of water and methanol increases, particles are formed as the drug precipitates (Appendix A). This can clinically be an obstruction to blood flow. Therefore, the composition ratio of the mixed solution of water and methanol is important.

In order to confirm the drug-coated porous PCL scaffold, Fourier transform infrared (FT-IR) experiments were performed; the results are shown in Figure 4. The spectrum of the PCL contains peaks at 2948 cm^−1^ (asymmetric CH_2_ stretching), 2860 cm^−1^ (symmetric CH_2_ stretching), 1730 cm^−1^ (carbonyl stretching), 1290 cm^−1^ (C–O and C–C stretching) and 1240 cm^−1^ (asymmetric COC stretching), respectively [37,38,39,40]. For drug-coated sample (blue line) and after drug-released sample (green line) characteristic absorption bands attributed to drugs were observed at 1540 cm^−1^. Then, at around 1540 cm^−1^, one peak was visible only in the drug-coated porous PCL (blue and green line) which was attributed to the N–H bending vibrations of the atorvastatin calcium salt. The presence of an amide group in the FT-IR spectrum of the drug-coated porous PCL scaffolds indicates that the drug (atorvastatin calcium salt) was attached to the PCL surface. In addition, the N–H bending peak was decreased after the drug release experiment. It is evident from the corresponding peaks of the drug-coated porous PCL scaffold on the FT-IR spectra that the drug was incorporated onto the surface of the porous PCL scaffold. Furthermore, the amount of aspirin: atorvastatin calcium salt in the coatings was confirmed by an elution test. A comparison of the aspirin and atorvastatin calcium salt is presented in Appendix A (drug content).

To confirm the drug coating on the porous PCL scaffold, surface characterization was done with XPS. On the survey scan spectra, the high resolution C_1s_ and O_1s_ peaks of the porous PCL scaffold and drug-coated porous PCL scaffold are shown in Figure 5. Two separated peaks corresponding to C_1s_ (284 eV) and O_1s_ (532 eV) are seen in all the XPS spectra (Figure 5) [40,41,42,43]. The surface chemical composition of the porous PCL scaffold was 76.55% carbon and 23.45% oxygen. After the drug coating on the porous PCL scaffold, the surface composition was 73.99% carbon and 25.55% oxygen. In the case of the drug-coated porous PCL sample, the content of both oxygen and carbon was increased compared to the sample without the drug coating. The difference between the porous PCL and drug-coated porous PCL scaffold was attributed to the use of the drug. A distinct N_1s_ peak at 400 eV was observed in the spectra of the drug-coated porous PCL scaffold (Figure 5d), confirming the successful introduction of N–H groups into the porous PCL scaffold. The nitrogen surface concentration increased from 0% to 0.25% after the drug coating (Table 1) with the additional nitrogen present from the atorvastatin calcium salt. The presence of the drug was further detected based on a Ca_2p_ peak at 347 eV observed only in the spectrum of the drug-coated porous PCL scaffold (Appendix A) with calcium ions present from the atorvastatin calcium salt.

## 4. Conclusions

In summary, this paper presented the fabrication and characterization of a drug-coated porous scaffold based on a bioabsorbable polymer, PCL, using a 3D printing system and a dip coating process. New bioresorbable drug-coated vascular scaffolds are currently being extensively developed. Qualitative and quantitative chemical analyses should be done for the drugs in scaffolds. A facile technique was used to obtain drug-coated scaffolds by modifying the surface. Our porous PCL scaffold was fabricated with a 3D printing system, and its surface was converted to a hydrophilic surface using LF plasma, and the aspirin and atorvastatin calcium salt mixture was dip coated onto the surface using an ultrasonic homogenizer. Our drug-coated porous PCL scaffold could be used to treat carotid artery stenosis. The drug-coated porous PCL scaffold produced from a bioresorbable polymer is cost-effective and has good compatibility with blood; hence, it could be used as an effective platform for biomedical application research. Furthermore, it could serve as a better option for a stent compared to a metallic stent.

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
