# Peer review of "Fabrication and Characterization of Bioresorbable Drug-coated Porous Scaffolds for Vascular Tissue Engineering"

_materials, 2019, doi:10.3390/ma12091438_

Reviewer 1 Report

Title: Fabrication and Characterization of Bioabsorbable Drug-coated Porous Scaffolds for Vascular Tissue Engineering

Very interesting research paper however, there is a need for the following corrections:

Minor Corrections:

-         Abstract is too generic specifically the materials and method section as well as result. There is not enough information in abstract about which drug used, which technique for coating drug on surface, no numerical data as the optimum results and absolutely no results shown in abstract.

-         Abstract: English need to be revised slightly.

-         Author can add the following sentence to the abstract: “Fabricated graft was coated with aspirin and atorvastatin calcium salt to reduce the blood LDL cholesterol and restenosis”.

-         Author mentioned that “A stainless barrel was used to contain the mixture for the 3D printing at 60 °C”. There is need for more information in this section of materials and methods.

-         Do not use we in your sentences. All manuscript needs to be written scientifically.

-         Figure 2 – What is the magnification of SEM and optical images? You mentioned about scale bar which is great but it would be useful to add magnification as well.

-         What is figure S1 and S2. I have not seen these data.

-         Where is HPLC results?

-         Need more discussion about Figure 4 and 5. Very short statements provided and not critically analysied/discussed.

Overall, good paper but need to do above correction before it can be accepted.

Author Response

We appreciate your kind comments. We apologize the confused expressions.

They were correctly revised in the paper as color fonts.

As shown below,

-         Abstract is too generic specifically the materials and method section as well as result. There is not enough information in abstract about which drug used, which technique for coating drug on surface, no numerical data as the optimum results and absolutely no results shown in abstract.

☞ We have modified the abstract according to your kind comment. We described the type of drug we used, the coating method, and the expected effect.

-         Abstract: English need to be revised slightly.

☞ The unnecessary parts were modified. Nevertheless, if you recommend an English proofreading, we'll leave later in the English correctional institutions.

-         Author can add the following sentence to the abstract: “Fabricated graft was coated with aspirin and atorvastatin calcium salt to reduce the blood LDL cholesterol and restenosis”.

☞ We thank you for your good comment and we have added the above sentence to the abstract.

-         Author mentioned that “A stainless barrel was used to contain the mixture for the 3D printing at 60 °C”. There is need for more information in this section of materials and methods.

☞ In response to your question, we have specifically described the materials and methods.

-         Do not use we in your sentences. All manuscript needs to be written scientifically.

☞ Based on your good suggestions, we refrained from using we in the sentences, except when we experimented directly.

-         Figure 2 – What is the magnification of SEM and optical images? You mentioned about scale bar which is great but it would be useful to add magnification as well.

☞ It was my mistake perfectly. To clarify the exact magnification, a new SEM and optical microscope images were taken and the Figures 2 and Figure 3 were completely corrected.

-         What is figure S1 and S2. I have not seen these data.

☞ Figures S1 and S2 are in the supplementary information and have been revised once again.

-         Where is HPLC results?

☞ The HPLC data was inserted into the supplementary information in Figure S3.

-         Need more discussion about Figure 4 and 5. Very short statements provided and not critically analysied/discussed.

☞ We respect your comment and added analysis and explanation to Figures 4 and 5.

Reviewer 2 Report

A very interesting works on the production of Bioabsorbable Drug-coated Porous Scaffolds for Vascular Tissue Engineering with 3D printing method, together with characterisation using FTIR, different type of microscopy and spectrophotometry analysis- I enjoy reading the work. However, to test if the materials do support vascular formation it would be more useful to apply some in vitro cell culture analysis - normal or cancer cell line would do. This could answer if the drugs could support or inhibit the growth of cells cultivated, hence could also helps on understanding how to promote vascularisation. Overall content on material fabrication and characterisation is good with a clear flow of writing on the step and procedures, however, it needs more analysis on cell culture works to support the objective of this study.

Author Response

A very interesting works on the production of Bioabsorbable Drug-coated Porous Scaffolds for Vascular Tissue Engineering with 3D printing method, together with characterisation using FTIR, different type of microscopy and spectrophotometry analysis- I enjoy reading the work. However, to test if the materials do support vascular formation it would be more useful to apply some in vitro cell culture analysis - normal or cancer cell line would do. This could answer if the drugs could support or inhibit the growth of cells cultivated, hence could also helps on understanding how to promote vascularisation. Overall content on material fabrication and characterisation is good with a clear flow of writing on the step and procedures, however, it needs more analysis on cell culture works to support the objective of this study.

☞ We appreciate your kind comments. As your comments, in vitro analysis is more useful for vascular scaffold fabrication. Actually, this paper was to study drug coating technology for biodegradable scaffold application. Therefore, this paper focused on the evaluation of drug elution using scaffold made by 3D printing. We will try not only in vitro test but also in vivo study in the future study.

Reviewer 3 Report

Dear editor,

The manuscript “Fabrication and Characterization of Bioabsorbable 2 Drug-coated Porous Scaffolds for Vascular Tissue 3 Engineering” describe the preparation of porous PCL scaffold using 3D printing. Materials were further modified (coated) by Atorvastatin Calcium Salt/Aspirin and characterised for morphology, and drug content. Manuscript is written logically and systematically; understandable English. However, some issues should be addressed before publication. Therefore, I have the following comments on the submitted manuscript:

General comments:

·        The used term bioabsorbable is rarely used, I find it more appropriate to use the term bioresorbable instead (more commonly referred to in the literature).

·        There is a complete lack of drug release characteristics - I think it is an essential part of the work focused on preparation of drug-delivery material.

ABSTRACT

·        Abstract should give more space to content of own work incl. achieved results.

·        Page1, line 25/26 : …. “PCL scaffolds with biocompatibility”…..this expression is not correct

INTRODUCTION

·        Page 2/line 52 : ….”PCL is one of the most elastic among the synthetic bioabsorbable polymers” …. The source (literature) for this claim should be added

MATERIALS and METHODS

·        the authors describe Sigma Aldrich as the source of all chemicals, but other sources are mentioned below - it is misleading.

·        Page 3/line 85 ….indicate the percentage composition of DMSO solution

RESULTS and Discussions

·        Page 4/line140: the claim that it is a hydrophilic surface (PCL surface) should be experimentally verified

·        Page5/line153-160: the exact composition of the drug solutions (water: methanol ratio) should be indicated

·        Fig.3. : the picture of negative control (drug-uncoated material) should be added to this fig – at the comparable magnification). Furthermore, it would be useful to unify the magnifications or scales in the Fig.2 and Fig.3.

·        Fig.4.: the image lacks identification of samples a,b and c.

For the above reasons, I propose to revise the article.

Author Response

We appreciate your kind comments. We apologize the confused expressions.

 They were correctly revised in the paper as color fonts.

As shown below,

General comments:

·        The used term bioabsorbable is rarely used, I find it more appropriate to use the term

          bioresorbable instead (more commonly referred to in the literature).

☞ We respect your comments and have made changes to your proposed term.

·        There is a complete lack of drug release characteristics - I think it is an essential part of the

          work focused on preparation of drug-delivery material.

☞ We have modified the abstract according to your kind comment. So, we focused on preparation of drug-coated material.

ABSTRACT

·        Abstract should give more space to content of own work incl. achieved results.

☞ We revised the abstract according to your valuable comments. And, we described the type of drug we used, the coating method, and the expected effect.

·        Page1, line 25/26 : …. “PCL scaffolds with biocompatibility”…..this expression is not correct

☞ We corrected the inaccurate expression.

INTRODUCTION

·        Page 2/line 52 : ….”PCL is one of the most elastic among the synthetic bioabsorbable polymers” …. The source (literature) for this claim should be added

☞ We have added references to support the rationale for the content. (Ref. 14-16)

MATERIALS and METHODS

·        the authors describe Sigma Aldrich as the source of all chemicals, but other sources are mentioned below - it is misleading.

☞ We have specified each source of chemicals.

·        Page 3/line 85 ….indicate the percentage composition of DMSO solution

☞ Based on your question, we have specified the composition of the DMSO solution. (PCL:DMSO=60:40 volume ratio)

RESULTS and Discussions

·        Page 4/line140: the claim that it is a hydrophilic surface (PCL surface) should be experimentally verified

☞ Respecting your comments, we have confirmed the hydrophilicity characteristics through PCL surface treatment experiments. Plasma treatment of the PCL surface confirmed that it changed from hydrophobic to hydrophilic. This result was inserted into supplementary information (Figure S1).

·        Page5/line153-160: the exact composition of the drug solutions (water: methanol ratio) should be indicated

☞ Based on your question, we have specified the composition of the drug solution. (Water:MeOH=1:1 volume ratio)

·        Fig.3. : the picture of negative control (drug-uncoated material) should be added to this fig – at the comparable magnification). Furthermore, it would be useful to unify the magnifications or scales in the Fig.2 and Fig.3.

☞ It was my mistake perfectly. To clarify the exact magnification, a new SEM and optical microscope images were taken and the Figures 2 and Figure 3 were completely corrected.

·        Fig.4.: the image lacks identification of samples a,b and c.

☞ We respect your comment and added analysis and explanation to Figure 4.

Round  2

Reviewer 2 Report

I still believe that it is needed for in vitro cell analysis as per topic discussed on vascular tissue engineering. Biocompatibility of drug-coated polymer used is still needed to be further tested with cells in vitro. 

Author Response

I still believe that it is needed for in vitro cell analysis as per topic discussed on vascular tissue engineering. Biocompatibility of drug-coated polymer used is still needed to be further tested with cells in vitro. 

☞ We appreciate your kind comments.

PCL is a semi-crystalline, linear, resorbable, aliphatic polyester approved by the U.S. Food and Drug Administration (FDA) for use in medical devices. Because of its biocompatibility, this polymer has been studied for use as tissue engineering and medical applications [Ref.21]. We described the above in the introduction and used PCL. And we focused on the preparation of drug-coated material. Therefore, we will try in vitro cell analysis in the future study.

Reviewer 3 Report

dear authors,

I have only one comment on the modified manuscript - it is fig. 2. The label below the scale picture does not match (there are two different scales for a).

Author Response

We appreciate your kind comments. We apologize the confused expressions.

They were correctly revised in the paper as color fonts.

As shown below,

I have only one comment on the modified manuscript - it is fig. 2. The label below the scale picture does not match (there are two different scales for a).

☞ It was my mistake, we have revised what you mentioned.

Figure 2. (a) Optical microscopic image of the porous PCL scaffold. (b, c) Top-view SEM image of the porous PCL scaffold. Scale bars: 500 mm (a), 200 mm (b) and 10 mm (c).